# Approaches to Testing Novel β-Lactam and β-Lactam Combination Agents in the Clinical Laboratory

**DOI:** 10.3390/antibiotics12121700

**Published:** 2023-12-05

**Authors:** Carmella Russo, Romney Humphries

**Affiliations:** Pathology, Microbiology and Immunology, Vanderbilt University Medical Center, Nashville, TN 37232, USA; cerusso27@gmail.com

**Keywords:** antimicrobial susceptibility testing, breakpoints, beta-lactams, beta-lactam combination

## Abstract

The rapid emergence of multi-drug resistant Gram-negative pathogens has driven the introduction of novel β-lactam combination agents (BLCs) to the antibiotic market: ceftolozane-tazobactam, ceftazidime-avibactam, meropenem-vaborbactam, imipenem-relebactam, cefiderocol, and sulbactam-durlobactam. These agents are equipped with innovative mechanisms that confer broad Gram-negative activity, notably against certain challenging carbapenemases. While their introduction offers a beacon of hope, clinical microbiology laboratories must navigate the complexities of susceptibility testing for these agents due to their diverse activity profiles against specific β-lactamases and the possibility of acquired resistance mechanisms in some bacterial isolates. This review explores the complexities of these novel antimicrobial agents detailing the intricacies of their application, providing guidance on the nuances of susceptibility testing, interpretation, and result reporting in clinical microbiology laboratories.

## 1. Introduction

The global spread of multi-drug and extensively resistant Gram-negative bacteria is a serious threat to public health [1]. Fortunately, several antimicrobials that target emerging resistance mechanisms have become available. These include novel β-lactam combination (BLC) agents that combine a β-lactam with a β-lactamase inhibitors, as well as the novel cephalosporin, cefiderocol [2,3]. However, susceptibility to these agents is not universal, and resistance is encountered in both patients treatment-naïve to [4,5] or exposed to these agents [2]. As such, accurate testing by the clinical laboratory of Gram-negative bacteria against these newer antimicrobials is critical. In this review, we discuss testing for novel β-lactam and BLC agents with indications for Gram-negative bacteria that are currently marketed in the United States. We describe when to test, what to expect results wise, and how to evaluate unusual resistance profiles for these agents.

## 2. Ceftolozane-Tazobactam

Ceftolozane-tazobactam (C/T) combines tazobactam, a penicillinase sulfone irreversible β-lactamase inhibitor, with ceftolozane, an engineered expanded-spectrum cephalosporin. Ceftolozane was designed to improve on the chemical properties of ceftazidime, increasing membrane permeability and stability against class C β-lactamases, including the chromosomally encoded *Pseudomonas*-derived cephalosporinase (PDC) [6]. Tazobactam provides the protection of ceftolozane to selected β-lactamases, including some expanded spectrum β-lactamases (ESBL), expanding activity in Enterobacterales.

C/T demonstrates a broad activity against common Gram-negative bacteria and in particular multidrug-resistant *P. aeruginosa*. Multiple studies have shown 80–98% of all *P. aeruginosa* are susceptible to C/T, including 68–89% of multi-drug resistant (MDR) strains [7,8]. Activity among Enterobacterales is high (>90%) and 88% of ESBL-producing Enterobacterales are susceptible [9]. C/T demonstrates no activity against carbapenemase-producing isolates, including the most common carbapenem-resistant Enterobacterales (CRE) (Table 1). In contrast, C/T may demonstrate activity against carbapenem-resistant *P. aeruginosa*, where the resistance mechanism is less commonly attributed to carbapenemase expression [10]. However, PDC variants resistant to C/T have emerged in clinical isolates of *P. aeruginosa* [11,12]. Furthermore, PDC overexpression in *P. aeruginosa* leads to low-level C/T resistance [13].

### 2.1. Testing Ceftolozane-Tazobactam in the Clinical Laboratory

C/T is often first-line therapy for MDR and difficult-to-treat (DTR) *P. aeruginosa* [14] and testing such isolates is indicated. Requests to test isolates of Enterobacterales are less frequent, but C/T might be considered by clinicians in cases of mixed infections with *P. aeruginosa*, or as a carbapenem-sparing option for an ESBL-producing Enterobacterales. Testing should be performed in these instances to ensure the appropriateness of C/T as a treatment option and to monitor the susceptibility of the organisms to this combination. C/T testing is performed using a fixed concentration (4 µg/mL) of tazobactam [15].

The US Food and Drug Administration (FDA) [16], the Clinical and Laboratory Standards Institute (CLSI) [15], and the European Committee for Antimicrobial Susceptibility Testing (EUCAST) [17] publish Enterobacterales and *P. aeruginosa* breakpoints for C/T. Breakpoints differ between CLSI/FDA and EUCAST for these organisms (Table 2). CLSI published breakpoints for *Haemophilus influenzae* and viridans group *Streptococcus* are recognized by the FDA, and the FDA lists breakpoints for *Bacteroides fragilis*. EUCAST breakpoints include an area of technical uncertainty for Enterobacterales, meaning isolates with zones of growth inhibition surrounding the C/T disk of 19–21 mm may be susceptible by MIC, despite these zones being considered resistant by the EUCAST breakpoints [17].

FDA-cleared methods are summarized in Table 3. In general, C/T is available on all commercially available platforms and is routinely tested by many laboratories. The performance of these tests is generally good, although issues with the EUCAST disk correlates (breakpoints) have been noted [18,19,20,21]. When first released, gradient diffusion strips were reported to perform sub-optimally for *P. aeruginosa*, but these findings could not be repeated when the correct formulation of antimicrobial was used as a comparator [22].

### 2.2. Expected Testing Results

C/T resistance among Enterobacterales, particularly those that express an ESBL, is often observed. ESBLs most commonly associated with resistance to C/T include GES (Guiana extended spectrum beta-lactamase), PER (*Pseudomonas* extended resistant) and BEL (from Belgium) types [23]. Many MDR *P. aeruginosa* are susceptible to C/T; resistant to both ceftazidime and imipenem in *P. aeruginosa* often comes along with resistance to C/T [13]. Several studies have evaluated the differences in susceptibility between C/T and ceftazidime-avibactam (CZA) for *P. aeruginosa* and found slightly higher rates of susceptibility for C/T, although resistance rates in general are not dramatically different [24]. Isolates susceptible to ceftazidime but resistant to C/T should be re-tested, as this indicates a testing error; advice for this is given below and in Table 4. If isolates positive for the carbapenemase gene or carbapenamase activity test susceptible to C/T, consideration should be given to editing this result to resistant (Table 1).

## 3. Ceftazidime-Avibactam

Ceftazidime-avibactam (CZA) is the combination of the expanded spectrum cephalosporin, ceftazidime, and avibactam, a non-β-lactam β-lactamase inhibitor [2,25,26]. Avibactam acts specifically via the covalent acylation of β-lactamases in a process that is slowly reversible and results in the release of intact avibactam [26]. Unlike some β-lactamase inhibitors like clavulanate, avibactam does not induce chromosomal AmpC, making it useful against organisms carrying resistance via AmpC [25]. 

CZA demonstrates a broad spectrum of activity against a variety of β-lactamases expressed by Enterobacterales, including class A extended-spectrum β-lactamases (ESBLs), and serine carbapenemases like KPC, class C cephalosporinases, and some class D oxacillinases (Table 1). However, CZA does not inhibit the activity of metallo-β-lactamases (MBLs, such as NDM, IMP, or VIM). CZA is effective against *P. aeruginosa*, including some isolates resistant to other antipseudomonal β-lactams. However, as the majority of carbapenem resistance in *P. aeruginosa* is driven by the down-regulation of membrane porin expression and the upregulation of efflux systems, not transferable carbapenemases [10,27,28,29], the activity of CZA against carbapenem-resistant *P. aeruginosa* [24,30] is less than it is against CRE that do not express MBLs [31]. 

Non-MBL mediated CZA resistance in CRE is generally uncommon. However, isolates with a mutation to the *bla_KPC_* gene omega loop [32,33,34] and those with increased expression of *bla_KPC_* in the context of reduced permeability [35,36,37] may demonstrate CZA resistance. In the former case, the carbapenems (meropenem, imipenem, and ertapenem) often test susceptible due to the loss of carbapenemase activity in these KPC variants, leading to an unusual pattern of CZA resistance and meropenem susceptibility [38]. However, data show these isolates may re-gain carbapenem resistance through adaptive mutations and treatment with a carbapenem is generally not recommended [39]. Certain variants of AmpC (e.g., CMY-185) are also associated with CZA resistance in the absence of an MBL [40]. In addition, modification of PBP3 has been shown to lead to resistance to CZA [41,42,43].

### 3.1. Testing Ceftazidime-Avibactam Susceptibility in the Clinical Laboratory

Laboratories should consider testing for CZA susceptibility among isolates of CRE and *P. aeruginosa* that are resistant to anti-pseudomonal β-lactams. The Infectious Diseases Society of America (IDSA) recommends CZA for infections caused by CRE when carbapenemase testing results are not available or when the infecting isolate is positive for KPC or OXA-48 like carbapenemases [14]. For Enterobacterales positive for MBLs, a combination of CZA with aztreonam is suggested for treatment. Testing for this combination is discussed further below. CZA is recommended for the treatment of multidrug-resistant *P. aeruginosa* or extensively drug-resistant *P. aeruginosa*, provided they do not produce MBLs [14].

The FDA [16], CLSI [15] and EUCAST [17] all publish CZA breakpoints for Enterobacterales and *P. aeruginosa* (Table 2). MIC breakpoints are harmonized. CLSI and EUCAST use different disk masses—i.e., 30 µg of ceftazidime and 20 µg of avibactam is used in the CLSI/FDA disk and only 10 µg of ceftazidime and 4 µg of avibactam is used in the EUCAST disk—as a result, disk breakpoints differ [15,17].

CZA is now available on most FDA-cleared platforms used for routine antimicrobial susceptibility testing (AST) of Gram-negative bacteria (Table 3). Laboratories that encounter CRE with any frequency should consider routine testing and reporting for carbapenem-resistant isolates or those that express a class A or D carbapenemase, as data suggest early treatment with an active antimicrobial can significantly reduce mortality associated with these infections [44]. Of note, there is no intermediate breakpoint for CZA, which increases the risk of false-resistance and false-susceptibility for isolates with MICs at the breakpoint (i.e., there is no buffer zone). If the disk diffusion method is used, CLSI recommends confirming Enterobacterales isolates with zone of growth inhibition diameters of 20–22 mm (i.e., at the susceptible/resistant breakpoint) by an alternate method to avoid reporting false-resistance or susceptibility. In contrast, the EUCAST defines *P. aeruginosa* zones of 16–17 mm as falling in the area of technical uncertainty—i.e., requiring confirmation. Overall performance for all commercially available methods is good [20,45,46].

### 3.2. Expected Testing Results

In most regions, CZA resistance is rare among Enterobacterales and *P. aeruginosa* [47]. Exceptions include regions with high rates of MBLs, such as Southeast Asia [48] or for multi-drug resistant *P. aeruginosa,* as described above. Based on local epidemiology, laboratories should consider confirming any CZA resistant results in Enterobacterales that are negative for an MBL. Carbapenem and CZA resistance is often indicative of the presence of an MBL, and this supposition should be confirmed, for treatment [14] and epidemiological/infection control purposes. CZA resistance in the context of carbapenem susceptibility should always be confirmed, and testing isolates for *bla*_KPC_ considered, given the risk of KPC variants causing this phenotype. For *P. aeruginosa*, CZA resistance may occur at higher rates, but generally in the context of MDR or DTR strains.

### 3.3. Combining CZA with Aztreonam

The combination of CZA with aztreonam has been shown to be an effective option for the treatment of isolates that express MBLs, including *S. maltophilia* [14]. The basis for this is that MBLs have no activity against aztreonam, and the avibactam component protects aztreonam from degradation by any concomitantly expressed β-lactamases, including Ambler class A, C and D enzymes [49]. Aztreonam has activity towards penicillin-binding protein 3 (PBP3), and isolates with the insertion YRIN/K at position 333 have reduced susceptibility to the aztreonam-avibactam combination; when combined with expression of some plasmid-borne CMY-type AmpC β-lactamases, these PBP3 mutations can lead to resistance to aztreonam-avibactam [50]. The combination aztreonam-avibactam has completed clinical trials but is not yet approved by the US FDA or the European Medicines Agency (EMA), and no commercial tests are available. However, testing the combination of CZA and aztreonam is possible through a variety of methods [51], including the broth-disk elution method endorsed by CLSI [52].

## 4. Meropenem-Vaborbactam

Meropenem-vaborbactam (MEV) consists of the synthetic carbapenem, meropenem, and a cyclic boronic acid β-lactamase inhibitor, vaborbactam. Vaborbactam extends the activity of meropenem against Gram-negative bacteria producing class A β-lactamases and class C cephalosporinases but lacks activity against MBLs and oxacillinases with carbapenemase activity, like OXA-48 [53]. Activity extends to some KPC variants that are resistant to CZA [54]. Resistance to MEV among KPC-producers has been observed, generally associated with reduced permeability due to porin mutation in combination with the overexpression of β-lactamase [55]. Importantly resistance to MEV due to the co-production of KPC with an OXA-48-like carbapenemase is common in certain regions [56].

### 4.1. Testing Meropenem-Vaborbactam in the Clinical Laboratory

Testing for MEV is primarily recommended for isolates of carbapenem-resistant Enterobacterales. The IDSA recommends MEV for infections caused by carbapenem-resistant Enterobacterales when carbapenemase testing results are unavailable, negative, or positive for KPC [14]. However, MEV does not add benefits beyond meropenem alone for isolates that are resistant to ertapenem but susceptible to meropenem and imipenem. Similarly, vaborbactam, which was designed for activity against KPC specifically, does not expand the activity of meropenem in isolates of *P. aeruginosa* or other Gram-negative bacteria, where KPC is an uncommon resistance mechanism. 

MEV MIC testing is performed using a fixed concentration of 8 µg/mL of vaborbactam. CLSI, EUCAST, and the FDA publish MEV breakpoints for the Enterobacterales alone (Table 1). Of note, the susceptible breakpoints for MEV are higher than those of meropenem alone, which are ≤1 µg/mL by CLSI breakpoints and ≤2 µg/mL by EUCAST breakpoints. This is because MEV doses, as listed in the prescribing information (4g q8 given over 3 h), maximize meropenem exposures. From a laboratory perspective, this may lead to scenarios where MEV is susceptible but meropenem is resistant—at the same MIC (4 µg/mL for CLSI/FDA and 8 µg/mL for EUCAST breakpoints). Laboratories should be aware this may indicate the presence of an oxacillinase such as OXA-48, which may limit the activity of MEV [57]. CLSI recently determined such strains should be reported as susceptible to MEV, despite the presence of an OXA-48 enzyme (CLSI June 2023 Agenda book). 

Testing methods (Table 3) are available for MEV but are relatively new to the market. Etest MICs do not correlate well with those determined by reference broth microdilution for *Proteus mirabilis* and Etest should not be used for this species [58]. 

### 4.2. Expected Testing Results

MEV resistance among Enterobacterales is rare, outside of regions with high rates of MBLs. As noted, isolates may test susceptible to MEV and resistant to meropenem, at the same MIC. Upon encountering resistance to MEV, confirming the presence or absence of a carbapenemase presence using molecular or phenotypic testing methods is important for treatment and infection control purposes. MEV resistance in the context of susceptibility to other carbapenems should not occur, and these results must be evaluated. 

## 5. Imipenem-Relebactam 

Imipenem-Relebactam (IMR) is comprised of imipenem (given with cilastatin, a renal dehydropeptidase-1 inhibitor) and relebactam, a non-β-lactam bicyclic diazabicyclooctane β-lactamase inhibitor. Relebactam inhibits Ambler class A (e.g., CTX-M, TEM, SHV, KPC) and class C β-lactamases (including PDC) but not MBLs and has limited activity against class D β-lactamases (e.g., OXA-48-like). Because of relebactam’s positively charged side chain, it is relatively resistant to efflux, providing additional activity against this commo resistance mechanism in *P. aeruginosa* [59]. It is important to note that *Proteus*, *Providencia*, and *Morganella* demonstrate intrinsically elevated imipenem MICs due to poor permeability of imipenem across the outer membrane [15]. 

Like the other BLCs discussed, IMR resistance among Enterobacterales may occur in the context of porin disruption, with or without increased expression of KPC [55] or AmpC. IMR appears to harbor activity against some isolates that express KPC variants resistant to CZA [33]. Isolates of *Serratia marcescens* expressing the SME serine carbapenemase are also resistant [60]. Isolates of *P. aeruginosa* that express VEB-type ESBLs or some GES type carbapenemases have been shown to be resistant to IMR [61]. In addition, OprD depletion may lead to IMR resistance in *P. aeruginosa,* due to reduced uptake of imipenem [62]. IMR is often active against isolates of *P. aeruginosa* that are resistant to C/T and CZA [63].

### 5.1. Testing Imipenem-Relebactam in the Clinical Laboratory

IDSA recommends IMR for treating infections caused by carbapenem-resistant Enterobacterales when carbapenemase testing results are unavailable, negative, or positive for KPC [14]. IMR does not provide a benefit for isolates resistant to ertapenem but susceptible to meropenem and imipenem. IMR is a preferred option for DTR *P. aeruginosa* infections. While many isolates of DTR *P. aeruginosa* that are resistant to C/T (which is available on many automated AST systems) are resistant to IMR, there are select isolates which may display susceptibility, so testing IMR is warranted in these cases [64].

The CLSI, EUCAST, and FDA IMR breakpoints are shown in Table 1, with notable differences between the organizations. In vitro testing involves a fixed relebactam concentration of 4 µg/mL. IMR Enterobacterales breakpoints exclude the *Morganellaceae* group, as intrinsic resistance to imipenem precludes activity of IMR in this group. While FDA breakpoints exist for *Acinetobacter baumannii* complex and *Haemophilus influenzae*, relebactam adds no additional activity to imipenem for these organisms, and testing is not routinely indicated if imipenem testing was performed.

FDA-cleared commercially available antibiotic susceptibility testing methods for IMR are in Table 3. Limited data are available for the performance of these tests, but one study evaluating reference broth microdilution vs. disk and gradient diffusion demonstrated good agreement for Etest, but categorical agreement of only 78% for the research-use-only disk evaluated, which is not currently available in the US [65]. 

### 5.2. Expected Testing Results

IMR resistance among Enterobacterales is rare outside of regions with high rates of MBLs. Most isolates of *P. aeruginosa* are anticipated to be susceptible to IMR, including some that test resistant to C/T and CZA. Upon encountering resistance to IMR, confirming the presence or absence of a carbapenemase using molecular or phenotypic testing methods should be performed. If carbapenemase testing is positive for Class B β-lactamase, IMR should be reported as resistant. Instances of imipenem susceptibility and IMR resistance should not occur and likely indicate testing error. Similarly, isolates that are susceptible to imipenem can be assumed to be susceptible to IMR.

## 6. Cefiderocol

Cefiderocol (FDC) is a novel siderophore cephalosporin antibiotic that exhibits a unique mechanism to penetrate bacterial cells. Siderophores are iron-chelating metallophores, a class of secondary metabolites produced by bacteria in order to scavenge metal ions form the environment [66]. FDC consists of a cephalosporin core conjugated to a catechol moiety which chelates iron [3]. This iron-FDC complex is actively ushered into the bacterial cell through iron-transport systems across the outer membrane. Once inside the periplasmic space, FDC targets penicillin-binding protein 3 (PBP3), disrupting cell wall synthesis which leads to bactericidal activity. FDC demonstrates stability against hydrolysis by various β-lactamase enzymes including ESBLs, AmpC, and carbapenemases. Resistance to FDC appears to be associated with three primary mechanisms: mutations to iron-uptake pathways, alterations of AmpC β-lactamases, and mutation to the PBP3 target. Several cases of treatment-emergent FDC resistance in Enterobacterales have been associated with functional alterations to the catecholate siderophore receptor gene, *cirA* [67,68,69,70]. Among isolates of *P. aeruginosa*, mutations to genes belonging to the TonB-dependent receptors associated with iron acquisition, such as PirA and PiuA, have been shown to lead to FDC resistance [4,71]. Mutations to AmpC genes, particularly to the region encoding the R2 loop, are associated with elevated FDC MICs. These mutations are thought to widen the substrate binding site, thereby trapping FDC and limiting its activity [72]. Finally, mutation to PBP3 in the form of a 4- amino acid insertion at position 333, reduces the access of FDC to the PBP3 transpeptide pocket. Such mutations do not lead to outright resistance but are often found in isolates that express β-lactamases, such as NDM. Combined, these two mechanisms lead to FDC resistance [41]. Among many isolates, FDC resistance is more common when an MBL, particularly NDM, is expressed [73]. *S. maltophilia* isolates with MICs >1 µg/mL are rare [74].

### 6.1. Testing Cefiderocol in the Clinical Laboratory

The IDSA recommends reserving cefiderocol for treating infections caused by carbapenem-resistant organisms, including *Stenotrophomonas maltophilia* [14]. While breakpoints are available from CLSI, the FDA, and EUCAST (Table 2), there are limited testing options for clinical laboratories (Table 3), and those that are available are associated with significant limitations. 

FDC breakpoint evolution was complex, with CLSI publishing breakpoints prior to FDA and European Medicines Agency approvals for FDC, based on limited MIC and pharmacokinetic/pharmacodynamic data [75]. These breakpoints were updated once clinical data became available (Table 2). 

FDC testing may be desired for CRE and *P. aeruginosa*, particularly for MBL producers. Additionally, FDC is often the only treatment option for carbapenem-resistant *Acinetobacter* spp. or *S. maltophilia*, and testing may be indicated, particularly for *A. baumannii*. Testing, however, is complex. FDC broth-based testing requires an iron-depleted cation-adjusted Mueller Hinton broth (CA-MHB). Lot-to-lot variability in iron content for commercially prepared CA-MHB and iron present on supplies used to prepare MIC tests can lead to unacceptably high iron content, leading to artificially elevated FDC MICs [75,76]. The Sensititre FDC broth microdilution test was removed from market due to test inaccuracies, including falsely elevated MICs [77]. Disk diffusion is available, although again lot-to-lot variability in Mueller Hinton agar can lead to substantially inaccurate results, particularly when testing *Acinetobacter* spp. [76,77,78]. Work is ongoing with Shionogi to improve the tests, and it appears that careful attention to inoculum preparation, incubation times, and reading is key to ensuring accurate results [76]. 

### 6.2. Expected Testing Results

Most isolates should test susceptible to FDC, including isolates that express MBLs [79]. Resistance can emerge in therapy and repeat testing of isolates is warranted. Furthermore, resistance may be encountered in Enterobacterales that express NDM [80], those that express multiple carbapenemases and in *A. baumannii* isolates [79]. Resistance in *S. maltophilia* is exceedingly rare and should be confirmed by a reference method, if observed. Disk diffusion testing may lead to inaccurate results, and so careful review of the isolate’s overall phenotype is imperative. FDC resistance in the absence of resistance to other cephalosporins or carbapenems is unlikely and likely represents testing errors.

## 7. Sulbactam-Durlobactam

Sulbactam-dulrobactam (SUD) is designed to combat infections caused by *Acinetobacter* species, including those caused by multidrug-resistant strains. Sulbactam is a sulfone β-lactamase inhibitor that blocks essential PBPs of *Acinetobacter* and some other Gram-negative bacteria (e.g., *Neisseria* spp.), thereby disrupting bacterial cell wall synthesis which leads to bacterial cell lysis and death. Sulbactam does not display activity against Enterobacterales, *P. aeruginosa,* or most other Gram-negative bacteria. Durlobactam is a broad-spectrum serine β-lactamase inhibitor belonging to the diazabicyclooctane class of β-lactamase inhibitors. Durlobactam protects sulbactam from degradation by a wide range of β-lactamase enzymes, including (and unlike avibactam) the oxacillinases commonly found in *Acinetobacter* [81,82]. SUD exhibits limited activity against MBL-producing isolates as durlobactam does not inhibit these enzymes; however, MBL expression in *A. baumannii* is relatively uncommon to date. Resistance mechanisms outside MBLs are poorly defined to date, but appear to be related to a mutation of the PBP3 target of sulbactam and other β-lactamase variants not inhibited by durlobactam [82]. Importantly, such mutations do not translate to reduced susceptibility to other β-lactams, such as meropenem, imipenem or ceftazidime [83].

### Testing Sulbactam-Durlobactam in the Clinical Laboratory

Testing options include disks, gradient diffusion, and broth microdilution (Table 3). Very little data exist for the performance of these devices. Breakpoints are available from the FDA (Table 2). Most isolates should test susceptible to SUD, with >95% of carbapenem-resistant *A. baumannii* complex having MICs <4 µg/mL [84]. Resistance should be confirmed if encountered, particularly in isolates not expressing MBLs. As noted, SUD resistance may occur in the context of carbapenem or cephalosporin susceptibility, due to PBP3 mutation. 

## 8. Common Challenges When Testing Novel Βeta-Lactam Combination Agents and Cefiderocol

When clinical microbiology laboratories embark on AST of new BLCs, they encounter a myriad of hurdles, including understanding which organisms to test, access to commercially produced test systems, the use of appropriate breakpoints (interpretations), as well as troubleshooting unexpected results. These challenges call for comprehensive susceptibility testing, an understanding of the mechanisms of action and resistance, and continuous research to navigate the effective use of new β-lactam–β-lactamase inhibitor combinations in clinical practice.

### 8.1. Test Availability

Historically, it could take >5 years from the FDA approval of a new antimicrobial to the time when an AST becomes available to the clinical laboratory [85]. Today, this timeframe is abbreviated, but generally, manual tests (e.g., disk diffusion, gradient diffusion, or manual broth microdilution) are the only tests available in the first 1–3 years after the antimicrobial becomes available. Adding additional, manual tests to the laboratory is associated with increased workloads, quality control testing, and training requirements. As such, laboratories should coordinate closely with antimicrobial stewardship programs when determining which tests to bring in-house, and for which indications. While a given novel antimicrobial agent may achieve FDA clearance on automated systems, the release and marketing of these may be delayed by months or years as manufacturers work on software updates. Laboratories should routinely check in with the manufacturer of their AST on any forthcoming updates to their test system. 

### 8.2. Breakpoints

AST breakpoints (interpretive criteria) are available from a variety of organizations, including the U.S. FDA, CLSI and EUCAST as described throughout. Breakpoints for some agents may be published by one organization, but not all (as an example, CLSI and EUCAST do not have published SUD breakpoints at the time of this writing, whereas the FDA does). To complicate matters, breakpoints may be updated at different cadences by these organizations, leading to confusion on the part of the laboratory on which to use. As an example, FDC breakpoints were first published by CLSI, then the FDA, then updated by CLSI on two occasions [15,75]. Laboratories should make sure to review breakpoints in use in the laboratory and ensure that the most up to date standards are in use. Updating breakpoints on a test will require additional verification (if the test is FDA-cleared for the breakpoint) or validation (if the test is not cleared for the breakpoint); advice on this process is available from CLSI and the College of American Pathologists [86].

It is not infrequent that clinical teams desire testing an off-label organism against a new antimicrobial. For example, in the case of a multi-drug resistant *Achromobacter* or *S. maltophilia*. Disk diffusion should not be used in these cases, as disk criteria are organism-specific. MIC testing might be considered, after careful review of the organism’s intrinsic resistance patterns, available pharmacokinetic and pharmacodynamic data, and only after careful clinical consultation to discuss the limitations of this approach. Testing should only be undertaken using standard methods (e.g., use of Mueller Hinton agar and gradient diffusion or a reference broth microdilution method with standardized inoculum and incubation times); the alteration of methods for organisms that grow poorly may lead to inaccurate results.

### 8.3. Quality Assurance beyond Quality Control Testing

Quality control testing is an important component of the quality system used by laboratories to ensure the accuracy of testing. QC refers to testing a small panel of bacterial strains with known MIC or disk results using the AST device. QC strains are available from strain collections such as the American Type Culture Collection and have expected MIC and disk diffusion ranges available from CLSI and EUCAST. QC ranges are established as part of the drug’s development and reviewed for suitability by CLSI and EUCAST. Historically, this activity was performed individually, but CLSI and EUCAST now work jointly to ensure QC ranges are harmonized across both organizations. The laboratory must ensure that the test results are within the expected range for these QC organisms, as a quality measure for accurate results for patient care testing that is happening in parallel. The frequency of performing QC varies by laboratory and region, but at minimum, laboratories should perform QC testing weekly and upon a new lot or new shipment of reagents. 

While a valuable quality measure, QC testing is a blunt assessment of the test’s performance, detecting gross errors (such as, for example, a degradation of the test reagents). In practice, quality assurance also requires careful consideration of the pre-through-post analytical stages. Testing should be performed for the indicated organisms only, that display resistance to more narrow-spectrum options. Testing methods should be followed exactly, either by CLSI or EUCAST standards, or, for commercial methods, by following the manufacturer’s instructions for use. Careful review of each and every result must be performed before these are reported to the patient’s chart. Laboratory technologists must be able to identify unusual or unexpected results, which requires a thorough understanding of the antimicrobial agent, typical resistance mechanisms and local epidemiology.

### 8.4. Discrepancy in Testing Methods

Unusual or unexpected results, which are described above for each agent, should be investigated. Investigations should include careful scrutiny of the purity plate and original culture plate from which the test was set up. In our experience, mixed cultures may suggest contamination of the test not readily visible on the purity plate, repeating the test from a pure subculture may resolve the discrepant results. In addition, repeating the initial tests, including organism identification, will help rule out other random errors (such as an uninoculated well or a culture mix-up). If these actions do not resolve the unusual result, testing by orthologous methods, such as a disk diffusion test or gradient diffusion test, if available, may help provide clarity. It is important to be aware that all commercial test methods (including disk diffusion) are calibrated against a reference broth microdilution method—however, different testing methods may demonstrate bias in one direction or another, yielding discordant results from each other [87]. Such discrepancies are a significant hurdle at the bench. The laboratory may consider performing additional tests, such as molecular or enzymatic methods to detect the presence of a β-lactamase, to resolve the discrepancy [15]. As an example, evaluation of CZA resistance in Enterobacterales for a laboratory that rarely encounters such results could include: review of other β-lactam combinations like MEV and IMR, if tested to determine if these are concordant with the CZA result; review of the culture and purity plate to ensure no organisms are contaminating the test (e.g., *S. maltophilia*); evaluation for the presence of an MBL via PCR, lateral flow or phenotypic testing which would explain the resistance; repeating the test to rule out random errors; evaluation for the presence of KPC, which could be a variant resistant to CZA (usually this occurs in the context of meropenem susceptibility). If none of these investigations confirm the result, the laboratory may consider sending the isolate to a reference laboratory for further evaluation.

Several factors should be considered when pursuing these investigations. First, alerting the clinical team of the evaluation is generally warranted so they can consider the uncertain AST results when formulating treatment plans. Second, ensuring that only final, resolved results are included in the annual antibiogram, to avoid introducing bias to these data. Finally, the laboratory should take note of these instances, in order to identify trends.

### 8.5. Training

As mentioned, an in-depth knowledge of the AST system and the antimicrobial agent’s activity is needed to ensure accurate results. Increasingly, medical laboratory technologists work in laboratories that are understaffed. The testing of the agents described herein is complex and is needed for patients with infections associated with high mortality, due to few treatment options. Accurate test results are critical to the outcome for these patients. Demonstrated competency on the basic principles of the antimicrobial agent, expected test results, regional epidemiology, and testing limitations is required as part of training when implementing these tests into routine clinical laboratory test menus. The value of having an expert available, either a senior technologist, technical specialist, or microbiologist to consult on results cannot be under-emphasized. Careful scrutiny of results by clinical teams, including antimicrobial stewardship programs, will add a second layer to quality assurance for AST.

## 9. Conclusions

The new antimicrobials described herein mark a significant stride in antimicrobial therapy. These agents offer promising solutions against challenging infections, especially those caused by Enterobacterales and DTR-*P. aeruginosa*. Clinical microbiology laboratories play a pivotal role, tasked with ensuring accurate susceptibility testing for these agents. This endeavor demands a synergistic approach, combining the expertise of laboratory directors and antimicrobial stewardship teams. Adhering to the standards set by CLSI and EUCAST ensures consistency and reliability in the results. Whether opting for routine or on-demand testing, the presence of FDA-approved methodologies facilitates this process. 

## Figures and Tables

**Table 1 antibiotics-12-01700-t001:** Enzyme spectrum for novel beta-lactam combination agents.

	Enzyme	C/T	CZA	MEV	IMR
**Class A**	KPC	✗	✓	✓	✓
	SHV	✓	✓	✓	✓
	TEM	✓	✓	✓	✓
	CTX-M	✓	✓	✓	✓
**Class B (MBL)**	NDM	✗	✗	✗	✗
	IMP	✗	✗	✗	✗
	VIM	✗	✗	✗	✗
**Class C**	AmpC	✗	✓	✓	✓
**Class D**	OXA	✗	✓/✗	✗	✗

C/T, ceftolozane-tazobactam; CZA, ceftazidime-avibactam; MEV, meropenem-vaborbactam; IMR, imipenem-relebactam. ✗, no activity; ✓, activity; ✓/✗, partial activity.

**Table 2 antibiotics-12-01700-t002:** MIC breakpoints for novel beta-lactam agents.

Antimicrobial	Organism	CLSI			EUCAST			FDA		
		S	I	R	S	I	R	S	I	R
**CZA**	*Enterobacterales*	≤8/4	-	≥16/4	≤8/4	-	>8/4	Recognizes CLSI		
	*P. aeruginosa*	≤8/4	-	≥16/4	≤8/4	-	>8/4	Recognizes CLSI		
**C/T**	*Enterobacterales*	≤2/4	4/4	≥8/4	≤2	-	>2	Recognizes CLSI		
	*P. aeruginosa*	≤4/4	8/4	≥16/4	≤4	-	>4	Recognizes CLSI		
	*H. influenzae*	≤0.5/4	-	-	-	-	-	Recognizes CLSI		
	*Streptococcus, Viridans* Grp	≤8/4	16/4	≥32/4	-	-	-	Recognizes CLSI		
	*B. fragilis*	-	-	-	-	-	-	≤8/4	16/4	≥32/4
**MEV**	*Enterobacterales*	≤4/8	8/8	16/8	≤8/8	-	>8/8	Recognizes CLSI		
**IMR**	*Enterobacterales*	≤1/4	2/4	≥4/4	≤2/4	-	>2/4	Recognizes CLSI		
	*P. aeruginosa*	≤2/4	4/4	≥8/4	≤2/4	-	>2/4	Recognizes CLSI		
	*Acinetobacter calcoaceticus-baumannii* complex	-	-	-	≤2/4	-	>2/4	≤2/4	4/4	≥8/4
	*Haemophilus influenzae*	-	-	-	-	-	-	≤4/4		
	Anaerobes	≤4/4	8/4	≥16/4	-	-	-	Recognizes CLSI		
**FDC**	*Enterobacterales*	≤4	8	≥16	≤2	-	>2	Recognizes CLSI		
	*P. aeruginosa*	≤4	8	≥16	≤2	-	>2	≤1	2	≥4
	*Acinetobacter calcoaceticus-baumannii* complex	≤4	8	≥16	-	-	-	≤1	2	≥4
	*Stenotrophomonas maltophilia*	≤1	-	-	-	-	-	-	-	-
**SUD**	*Acinetobacter calcoaceticus-baumannii* complex	-	-	-	-	-	-	≤4/4	8	≥16/4

CLSI, Clinical and Laboratory Standards Institute; EUCAST, European Committee on Antimicrobial Susceptibility Testing; FDA, US Food and Drug Administration; S, susceptible; I, intermediate; R, resistant; C/T, ceftolozane-tazobactam; CZA, ceftazidime-avibactam; MEV, meropenem-vaborbactam; IMR, imipenem-relebactam; FDC, cefiderocol; SUD, sulbactam-durlobactam.

**Table 3 antibiotics-12-01700-t003:** Current FDA clearance status for the antimicrobial agents described in this review.

	C/T	CZA	MEV	IMR	FDC	SUD
Phoenix (Becton Dickenson)	Y	Y	Y	N	N	N
Vitek-2 (BioMerieux)	Y	Y	Y	Y	N	N
MicroScan (Beckman Coulter)	Y	Y	Y	N	N	N
Sensititre (Thermo Fisher Scientific)	Y	Y	Y	Y	Y	Y
Disk (Becton Dickenson)	N	Y	N	N	Y	N
Disk (Hardy Diagnostics)	Y	Y	Y	Y	Y	Y
Gradient diffusion (Etest, BioMerieux)	Y	Y	Y	Y	N	(RUO)
Gradient diffusion (MTS, Liofilchem)	Y	Y	Y	Y	Y	N
NGP (Selux Diagnostics)	N	Y	Y	Y	N	N
PhenoTest BC (Accelerate Diagnostics)	N	(RUO)	N	N	N	N

C/T, ceftolozane-tazobactam; CZA, ceftazidime-avibactam; MEV, meropenem-vaborbactam; IMR, imipenem-relebactam; FDC, cefiderocol; SUD, sulbactam-durlobactam, Y, available on platform; N, not available on platform; RUO, research use only.

**Table 4 antibiotics-12-01700-t004:** Unexpected results and troubleshooting when testing novel beta-lactam agents in the clinical laboratory.

Antimicrobial	Results	Possible Cause	Next Steps/Resolution
**C/T**	C/T–R Ceftazidime–S	Testing error Contamination	Repeat AST, check for mixed cultures, uninoculated wells, etc. Often indicates random contamination of test panel.
	C/T–S ESBL + Carbapenemase −	Expected result	No confirmatory testing necessary
	C/T–R ESBL + Carbapenemase −	Expected result	No confirmatory testing necessary
	C/T–S Carbapenemase +	C/T has no activity against carbapenemases, likely testing error. Less common—inactive carbapenemase	Confirm findings to rule out errors. Consider testing for carbapenemase function by performing mCIM or similar test.
**CZA**	CZA–R ESC–S	Testing error Contamination	Repeat AST, check for mixed cultures, uninoculated wells, etc. Often indicates random contamination of test panel.
	CZA–R Carbapenems–S	Testing error Contamination Carbapenemase variants such as KPC-161	Repeat AST, check for mixed cultures, uninoculated wells, etc. Often indicates random contamination of test panel. If result confirms, perform carbapenemase test and gene for mutations. Consider reporting carbapenems as R if carbapenemase present.
	CZA–R Serine carbapenemase +	Testing error Contamination Carbapenemase variants such as KPC-161	Repeat AST, check for mixed cultures, uninoculated wells, etc. Often indicates random contamination of test panel. Evaluate if CZA MIC or disk zone at breakpoint—due to absence of intermediate category, errors may occur. If result confirms, evaluate carbapenemase gene for mutations
	CZA–S MBL +	Testing error Less common—inactive/low expression of MBL	Confirm findings to rule out errors. Evaluate if CZA MIC or disk zone at breakpoint—due to absence of intermediate category, errors may occur. Consider testing for carbapenemase function by performing mCIM/eCIM or similar test. Some isolates may test at the susceptible breakpoint (MIC of 8 µg/mL). If testing confirms initial findings, consider reporting CZA as R.
**MEV**	MEV–S Meropenem–R at same MIC	Expected result due to breakpoints (Table 2) May indicate presence of low-level carbapenemase, such as OXA-48	Repeat AST, check for mixed cultures, uninoculated wells, etc. Often indicates random contamination of test panel. Evaluate for carbapenemase presence, by mCIM and/or genotypic method. If carbapenemase +, report both MEV result and carbapenemase, or consider reporting MEV as resistant. If carbapenemase -, report MEV as tested.
	MEV–R Carbapenem–S	Testing error Contamination	Repeat AST, check for mixed cultures, uninoculated wells, etc. Often indicates random contamination of test panel.
	MEV–R KPC+	Reduced permeability due to porin mutation combined with overexpression of KPC or co-production of KPC with an OXA-48-like carbapenemase	Repeat AST, check for mixed cultures, uninoculated wells, etc. Often indicates random contamination of test panel. Evaluate for presence of other carbapenemases by genotypic test. May be due to MBL or OXA-48-like enzyme. If results confirmed, report as tested.
**IMR**	IMR–R Carbapenem–S	Testing error Contamination	Repeat AST, check for mixed cultures, uninoculated wells, etc. Often indicates random contamination of test panel.
	IMR–R KPC+	Reduced permeability due to porin mutation combined with overexpression of KPC Co-expression of MBL *Proteus*, *Providencia* and *Morganella* grp expected phenotype.	Repeat AST, check for mixed cultures, uninoculated wells, etc. Often indicates random contamination of test panel. Evaluate for presence of other carbapenemases by genotypic test. If confirmed, report as tested.
**FDC**	FDC–R ESC–S	Testing error Mutations to iron-uptake pathway	Repeat AST, check for mixed cultures, uninoculated wells, etc. Often indicates random contamination of test panel. Ideally, send isolate to a reference laboratory for FDC testing.
	FDC–R by disk/S by MIC FDC–S by disk/R by MIC	Testing challenges with FDC may lead to discordant results.	Carefully evaluate adherence to AST protocol including inoculum preparation, incuation time, and reading instructions. Confirm iron depletion complete in CA-MHB. Try testing on a different lot/brand of MHA.
**SUD**	SUD–R ESC–S	Testing error β-lactamase variants not inhibited by durlobactam	Repeat AST, check for mixed cultures, uninoculated wells, etc. Often indicates random contamination of test panel.

C/T, ceftolozane-tazobactam; CZA, ceftazidime-avibactam; MEV, meropenem-vaborbactam; IMR, imipenem-relebactam; FDC, cefiderocol; SUD, sulbactam-durlobactam; ESC, expanded-spectrum cephalosporin; S, susceptible; R, resistant.

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
