# Peer review of "Approaches to Testing Novel β-Lactam and β-Lactam Combination Agents in the Clinical Laboratory"

_antibiotics, 2023, doi:10.3390/antibiotics12121700_

Round 1
Reviewer 1 Report
Comments and Suggestions for Authors
Dear Authors,
Your review entitled "Approaches to testing novel β-lactam and β-lactam combination agents in the clinical laboratory" has been reviewed,
This review deserves attention since it highlights a very important topic in the fields of both Microbiology and Public Health,
The Paper is well written in English, well designed, and well represented,
Unfortunately, I have several Major and Minor remarks regarding this work, kindly find my remarks below:
01- In the Abstract Section, Line 11, when Authors used the abbreviation (BLCs) is it for "β-lactamase inhibitor combination or β-lactam combination"?
02- In the Keyword section, Authors are invited to remove the name of Antibiotics, and to replace it by "β-lactam" and "β-lactam inhibitor combination".
03- In the Introduction section, Line 27, after "threat to public health" authors are invited to add the following article as a reference for this idea:
Reference 01: Antimicrobial resistance: A public health challenge (2015).
04- In the Whole manuscript, when Authors list the name of a bacterium, they are invited to put the full name followed by its abbreviation, example "in Line 46" Pseudomonas aeruginosa (P. aeruginosa), then they can use P. aeruginosa instead of Pseudomonas aeruginosa in other parts of the manuscript.
05- In the Whole manuscript Authors are invited to put the tables in a matter of being after the text when they talked about it.
06- Since Authors, talked in this Review, in the section number 6 "Cefiderocol", talked about siderophores, Authors are invited to talk a little bit about "Metallophores" and "Siderophores" Since they are numerous in bacteria and they play an important role in bacterial pathogensis, In addition these metallophores including siderophores can be the target of a new family of Antibiotics. They can use the following article as reference for this point:
Reference 02: Towards new antibiotics classes targeting bacterial metallophores
07- In the Page 11, Line 357, Authors are invited to put the title of the section 8, in bold.
Best Regards,
Author Response
01- In the Abstract Section, Line 11, when Authors used the abbreviation (BLCs) is it for "β-lactamase inhibitor combination or β-lactam combination"?
This has been edited to “beta-lactam combination” throughout.
02- In the Keyword section, Authors are invited to remove the name of Antibiotics, and to replace it by "β-lactam" and "β-lactam inhibitor combination".
Adjusted as suggested
03- In the Introduction section, Line 27, after "threat to public health" authors are invited to add the following article as a reference for this idea:
Reference 01: Antimicrobial resistance: A public health challenge (2015).
Added as suggested
04- In the Whole manuscript, when Authors list the name of a bacterium, they are invited to put the full name followed by its abbreviation, example "in Line 46" Pseudomonas aeruginosa (P. aeruginosa), then they can use P. aeruginosa instead of Pseudomonas aeruginosa in other parts of the manuscript.
Edited as suggested
05- In the Whole manuscript Authors are invited to put the tables in a matter of being after the text when they talked about it.
We are not sure how to correct this, we confirmed they were all inserted into text after the first mention, with the exception of Table 4 which is quite large.
06- Since Authors, talked in this Review, in the section number 6 "Cefiderocol", talked about siderophores, Authors are invited to talk a little bit about "Metallophores" and "Siderophores" Since they are numerous in bacteria and they play an important role in bacterial pathogensis, In addition these metallophores including siderophores can be the target of a new family of Antibiotics. They can use the following article as reference for this point:
Reference 02: Towards new antibiotics classes targeting bacterial metallophores
This is added to the revised manuscript, as suggested, "Siderophores are iron-chelating metallophores, a class of secondary metabolites produced by bacteria in order to scavenge metal ions form the environment [63]."
07- In the Page 11, Line 357, Authors are invited to put the title of the section 8, in bold.
Corrected.
Reviewer 2 Report
Comments and Suggestions for Authors
Despite a few number of concerns, this manuscript is of high public health relevance.
Areas of concern:
Introduction
The introduction is too short for a review article. The authors did not elaborate on the novel B-lactam combination agents (origin of their introduction to the antibiotic market, different types, their availability, accessibility, affordability and efficacy). If there exist other types in addition to those described in this review, the authors should explain the reason for their choice for these ones.
Line 37: There should be another heading introducing the various combination agents before section 2.
All the Tables (1-4) lack keys below to explain what some signs mean; for eg, signs like √, *, á¹», etc.
Lines 91, 158, 222, 268, and 323: Should be titled Interpretation of the Results of Susceptibility Testing
Lines 93-94: What do these abbreviations mean? Write them in full
Line 269: Delete ‘‘in’’ between ‘‘of’’ and ‘‘regions’’
Line 441: Delete ‘‘an’’ before ‘‘Enterobacterales’’
Line 150-151: A recommendation would have been reformulated concerning the following statement: ‘‘Of note, there is no intermediate breakpoint for CZA, which increases the risk of false-resistance or false-susceptibility for isolates with MICs near the breakpoint’’
References
Reference 11 (line 510): check this consistency.
Comments on the Quality of English LanguageApart from a few English errors, the review was well-written.
Author Response
The introduction is too short for a review article. The authors did not elaborate on the novel B-lactam combination agents (origin of their introduction to the antibiotic market, different types, their availability, accessibility, affordability and efficacy). If there exist other types in addition to those described in this review, the authors should explain the reason for their choice for these ones.
Given this article is one of a set that will include a discussion of the beta-lactams and beta-lactam inhibitors, we have not edited as suggested by the reviewer as any content we provide with be redundant to the reader.
Line 37: There should be another heading introducing the various combination agents before section 2.
We thank the reviewer for this suggestion, but have opted to not change this as we want to ensure the individual antibiotics are clearly visible.
All the Tables (1-4) lack keys below to explain what some signs mean; for eg, signs like √, *, á¹», etc.
We apologize these have been added.
Lines 91, 158, 222, 268, and 323: Should be titled Interpretation of the Results of Susceptibility Testing
Edited to “Expected Testing Results”
Lines 93-94: What do these abbreviations mean? Write them in full
Updated to, GES (Guiana extended spectrum beta-lactamase), PER (Pseudomonas extended resistant) and BEL (from Belgium) types
Line 269: Delete ‘‘in’’ between ‘‘of’’ and ‘‘regions’’
Edited as suggested.
Line 441: Delete ‘‘an’’ before ‘‘Enterobacterales’’
Edited as suggested.
Line 150-151: A recommendation would have been reformulated concerning the following statement: ‘‘Of note, there is no intermediate breakpoint for CZA, which increases the risk of false-resistance or false-susceptibility for isolates with MICs near the breakpoint’’
Of note, there is no intermediate breakpoint for CZA, which increases the risk of false-resistance and false-susceptibility for isolates with MICs at the breakpoint (i.e., there is no buffer zone).
References
Reference 11 (line 510): check this consistency.
This reference is as listed in PubMed.
Reviewer 3 Report
Comments and Suggestions for Authors
This work highlights the rise of multi-drug resistant Gram-negative pathogens, leading to the development of new β-lactam / β-lactamase inhibitor combinations (BLCs) in the antibiotic market.. However, clinical microbiology laboratories face challenges in testing the susceptibility of these agents due to their varying activity profiles against specific β-lactamases and the potential for acquired resistance in certain bacterial isolates. The review delves into the intricacies of these novel antimicrobial agents, offering guidance on their application and susceptibility testing interpretation in clinical microbiology laboratories. Below are several comments regarding this manuscript,
- Table 4, it would be more informative to provide references for the isolate variants that exhibit resistance to the beta-lactam agents listed in the table.
- Section 3. Ceftazidime-Avibactam. Numerous studies have reported that resistance to ceftazidime-avibactam in pathogens is mediated through modifications in Penicillin-binding proteins 3 (PBP3), as indicated by the following references. It would be beneficial to incorporate these cases into the review of ceftazidime-avibactam:
1. Zhang Y, Kashikar A, Brown CA, Denys G, Bush K. Unusual Escherichia coli PBP 3 insertion sequence identified from a collection of carbapenem-resistant Enterobacteriaceae tested in vitro with a combination of ceftazidime-, ceftaroline-, or aztreonam-avibactam. Antimicrob Agents Chemother 2017; 61:e00389-17.
2. Periasamy H, Joshi P, Palwe S, Shrivastava R, Bhagwat S, Patel M. High prevalence of Escherichia coli clinical isolates in India harbouring four amino acid inserts in PBP3 adversely impacting activity of aztreonam/avibactam. J Antimicrob Chemother 2020; 75:1650–1.
3. Poirel L, Ortiz de la Rosa JM, Sakaoglu Z, Kusaksizoglu A, Sadek M, Nordmann P. NDM-35-producing ST167 Escherichia coli highly resistant to β-lactams including cefiderocol. Antimicrob Agents Chemother 2022; 66:e0031122.
4. Wang Q, Jin L, Sun S, et al. Occurrence of high levels of cefiderocol resistance in carbapenem-resistant Escherichia coli before its approval in China: a report from China CRE-network. Microbiol Spectr 2022; 10:e0267021.
- Lane 179-183, it is advisable to reconsider the wording of the sentences concerning the reported cases of ceftazidime-avibactam resistance linked to PBP3 mutations.
Author Response
- Table 4, it would be more informative to provide references for the isolate variants that exhibit resistance to the beta-lactam agents listed in the table.
We have provided summary of the “flavors” of mechanisms behind unusual testing results, but this table is not intended to be an in-depth review of the resistance mechanisms; rather a troubleshooting guide for the laboratory. Laboratories would not confirm the variants in most cases, and new variants are continually being discovered. As such, we have left this table as is.
- Section 3. Ceftazidime-Avibactam. Numerous studies have reported that resistance to ceftazidime-avibactam in pathogens is mediated through modifications in Penicillin-binding proteins 3 (PBP3), as indicated by the following references. It would be beneficial to incorporate these cases into the review of ceftazidime-avibactam:
- Zhang Y, Kashikar A, Brown CA, Denys G, Bush K. Unusual Escherichia coli PBP 3 insertion sequence identified from a collection of carbapenem-resistant Enterobacteriaceae tested in vitro with a combination of ceftazidime-, ceftaroline-, or aztreonam-avibactam. Antimicrob Agents Chemother 2017; 61:e00389-17.
- Periasamy H, Joshi P, Palwe S, Shrivastava R, Bhagwat S, Patel M. High prevalence of Escherichia coli clinical isolates in India harbouring four amino acid inserts in PBP3 adversely impacting activity of aztreonam/avibactam. J Antimicrob Chemother 2020; 75:1650–1.
- Poirel L, Ortiz de la Rosa JM, Sakaoglu Z, Kusaksizoglu A, Sadek M, Nordmann P. NDM-35-producing ST167 Escherichia coli highly resistant to β-lactams including cefiderocol. Antimicrob Agents Chemother 2022; 66:e0031122.
- Wang Q, Jin L, Sun S, et al. Occurrence of high levels of cefiderocol resistance in carbapenem-resistant Escherichia coli before its approval in China: a report from China CRE-network. Microbiol Spectr 2022; 10:e0267021.
We thank the reviewer and have added these references.
- Lane 179-183, it is advisable to reconsider the wording of the sentences concerning the reported cases of ceftazidime-avibactam resistance linked to PBP3 mutations.
We are not entirely sure what the reviewer is alluding to. We have reviewed for English language, these seem to be correct. Please clarify and we are happy to revise.
Round 2
Reviewer 1 Report
Comments and Suggestions for Authors
Dear Authors,
Your revised version of this Review has been reviewed,
Thank you for the modifications you made,
The article is better for publication in its present form,
Best Regards,